# At what cycle threshold level are dogs able to detect SARS-CoV-2 in humans?

Víctor M. Vidal-Martínez[1]*, Juan Manuel Mancilla-Tapia[2], Lilia C. Soler-Jiménez [1], Iván Velázquez-Abunader[1], Matilde Jiménez-Coello[3], Antonio Ortega-Pacheco[3], David Hernández-Mena [4]

1 Aquatic Pathology Laboratory, Centro de Investigación y de Estudios Avanzados del Instituto Politécnico Nacional (CINVESTAV–IPN) Unidad Mérida, Mérida, Yucatán, México, 2 Canine Training Center Obi-K19, Hermosillo, Sonora, México, 3 Universidad Autónoma de Yucatán, Mérida, Yucatán, México, 4 Instituto de Biología, Universidad Nacional Autónoma de México (UNAM), México City, México

* vvidal@cinvestav.mx

**Data Availability Statement:** All relevant data are within the paper and its Supporting Information files.

## Abstract

Dogs can discriminate between people infected with SARS-CoV-2 from those uninfected, although their results vary depending on the settings in which they are exposed to infected individuals or samples of urine, sweat or saliva. This variability likely depends on the viral load of infected people, which may be closely associated with physiological changes in infected patients. Determining this viral load is challenging, and a practical approach is to use the cycle threshold (Ct) value of a RT-qPCR test. The hypothesis was that dogs should have a specific Ct range at which they could detect people infected with SARS-CoV-2. Therefore, the objective was to determine this Ct range. Sweat samples and epidemiological data were collected from 89 infected and 289 non-infected individuals at real life settings (e.g. health centers, offices, football fields). To determine each person's infection status, the Norgen Biotek kit for RT-qPCR was used; targeting the N1 and N2 regions of the SARS-CoV-2 nucleocapsid N gene. The performance of 11 trained dogs was evaluated on sweat samples of 379 individuals to determine their sensitivity and specificity (± 95% Confidence Intervals; CI) in detecting SARS-CoV-2 infections. Additionally, the SARS-CoV-2 viral load was calculated from Ct values using a reference curve, and the Ct range at which dogs showed optimal performance was determined. Six dogs exhibited a marginal performance, as their sensitivity 95% CI overlapped with the region of randomness (50%). The remaining five dogs demonstrated sensitivity values between 67% and 87%, with none of their 95% CIs overlapping the randomness region. Regarding specificity, three dogs showed values between 87% and 92%, while all other dogs exhibited values of ≥ 90%. Dogs demonstrated higher detection accuracy in a range of Ct values between 18.49 and 29.17 for the N1 region and between 24.07 and 26.69 for the N2 region of the SARS-CoV-2 nucleocapsid gene. Detection significantly decreases for Ct values greater than 30 or less than 16, indicating an optimal range in which dogs are most effective. These performance values concur well with those reported for commercial rapid antigen tests for detecting SARS-CoV-2. Consequently, it is considered that using properly trained animals could offer a viable option to supplement existing diagnostic methods, allowing for rapid diagnosis while optimizing time and

**Funding:** JMMT Funding acquisition, project supported by the Mexican Government (CONAHCyT) for training dogs in the bio-detection of SARS-CoV-2 at the OBI-K19 training center. Agreement Number: 000000000317533 Consejo Nacional de Humanidades, Ciencias y Tecnologías https://conahcyt.mx/ The funders had no role in study design, data collection and analysis, decision to publish, or preparation of the manuscript.

**Competing interests:** The authors have declared that no competing interests exist.

economic resources. Moreover, this approach is ecologically sustainable, as it generates less waste compared to the use of rapid tests, while continuing to confirm positive cases.

## Introduction

Dogs can detect volatile organic compounds (VOCs) in sweat, saliva and urine of humans infected with SARS-CoV-2 [1–4]. Their detection performance, however, varies significantly, with sensitivity values ranging from 68% to 98% and specificity values from 94% to 100% [1, 5, 6]. This variability could be related to the inherent sensitivity of dogs to detect VOCs along with the environmental distractions present in real-life settings such as airports and health centers [7], and their detection limits within specific viral concentration thresholds. These limits are associated with the number of viral copies in a person, which depends on the stage of infection. Importantly, detection dogs trained for viral diseases do not identify the viral genome directly but instead detect changes in VOCs produced by altered metabolism during infection [1, 2]. This suggests that dogs are most effective within certain viral copy ranges [1–4], making it crucial to understand the limits of their detection capabilities. However, determining the exact number of viral copies per individual is costly, time-consuming, and requires highly specialized infrastructure, posing a challenge for further research.

An indirect way to determine the number of viral copies of SARS-CoV-2 in a person is by using the cycle threshold (Ct) value of a RT-qPCR. A cycle threshold, also known as quantification cycle (Cq), is defined as the number of cycles required for the fluorescent signal of a RT-qPCR to exceed that of the background and crossing a threshold for positivity [8]. The use of Ct as a proxy of the number of viral copies has been controversial, especially in the case of SARS-CoV-2 [9–11]. However, for many clinicians and public health professionals it is still a useful indicator of the degree of infectiousness [8, 12–15]. Thus, one way to know the ideal range of viral copies in which dogs efficiently detect SARS-CoV-2 would be by evaluating the number of dogs reading the samples of the same person as positive (or negative) and comparing those results with the Ct values of his/her RT-qPCR tests. For example, if a person's sample is identified as positive by most of the dogs (e.g. 10 out of 11 dogs), and the person's RT-qPCR has a specific Ct value, it would be possible to determine the Ct value at which most of the dogs are able to detect that person. By repeating this process for hundreds of people, a range could be determined in which dogs perform well in detecting people infected with SARS-CoV-2. At the same time, it would provide information on the Ct values at which the dogs' detection capacity decreases. Likewise, dogs that perform best within optimal Ct ranges could be prioritized to improve bioscreening breeds.

Between August 2021 and February 2022, financial support was provided by the Mexican Government (CONACyT) for training dogs at the OBI-K19 training center in Hermosillo, Sonora, México [7]. For this first generation, four of the six dogs detected positive samples from patients with SARS-CoV-2, with sensitivity and specificity values significantly different from those obtained randomly in the field. These dogs were exposed to gauze impregnated with sweat or saliva from patients from Hermosillo for 1 min. We considered it promising that the performance of the dogs would improve by exposing them for a longer time to gauze impregnated with sweat or saliva from people with SARS-CoV-2. Therefore, in the field work for this study, the dogs were exposed for 5 minutes to gauze impregnated with axillary sweat from 379 people in Mérida, Yucatán, Mexico (90 positive samples and 289 negative samples) to evaluate the effectiveness of the SARS-CoV-2 biodetection training received by the dogs. A

second generation of SARS-CoV-2 detection dogs was trained during this time. Based on previous experience, it was anticipated that additional training time and the additional impregnation time of the samples would enhance sensitivity and specificity levels [7]. Thus, the first hypothesis was that the second generation would have better levels of sensitivity and specificity compared to the results from a previous published study [7]. The second hypothesis was that dogs would perform significantly better in bio-detection of SARS-CoV-2 within a specific range of Ct values, and that outside this Ct range, there would be considerable variability in bio-detection performance, with some dogs outperforming others.

For each of the 379 individuals, in addition to the sweat samples, nasopharyngeal swabs were obtained and RT-qPCR was conducted using the Norgen Biotek kit. This kit employs a technology and probes designed by the United States Government's Centers for Disease Control and Prevention (CDC), based on N1 and N2 primer/probe sets, targeting two distinct regions of the SARS-CoV-2 nucleocapsid gene [16, 17]. Thus, both the bio-detection results of the dogs and RT-qPCR results were available for each one of the 379 people. Therefore, the objectives of this paper were two-fold: 1) Determine the sensitivity and specificity of the second generation of dogs trained for the bio-detection of SARS-CoV-2 in Mexico; 2) Establish the optimal Ct range at which dogs can detect SARS-CoV-2, and which dogs perform better outside this Ct range.

## Materials and methods

### The dogs

In the second generation of dogs trained, 11 dogs out of 16 completed their training for bio-detection of SARS-CoV-2. The list and characteristics of these dogs are presented in Table 1. The dogs were trained at the Canine Training Center Obi-K19, Hermosillo, Sonora, México following the methodology proposed by Mancilla-Tapia and colleagues [7]. With exception of Krilling, Leia, Sam and Mike (which belonged to the first generation) none of the dogs had previous training, and never were exposed to samples of people infected with SARS-CoV-2, but had the instinct and performance to cope with the training phases. Three dogs (Ziggy, Rufus, and Lucky) acquired coccidiomycosis [18]. However, they were treated with itraconazole and fluconasol, and recovered well to complete their training. None of the dogs acquired SARS-CoV-2 during the study as indicated by negative antigen tests.

The pseudo-scent used for dog training was designed by the American company Bio Detection K9 (BDK9), which is a protein that mimics the scent of SARS-CoV-2. According to

**Table 1. Details of the 11 fully trained dogs exposed to sweat samples of SARS-CoV-2 positive and negative humans.**

| Name | Age (yrs) | Breed | Sex | Training | Origin / Specialty |
|------|-----------|-------|-----|----------|--------------------|
| Ziggy | 1 | Belgian Malinois | Female | Sweat | Home dog/Sportline dog |
| Mona | 1 | Belgian Malinois | Female | Sweat | Green dog/Working dog |
| Lucky | 1 | Belgian Malinois | Male | Sweat | Green dog/Working dog |
| Rex | 1 | Belgian Malinois | Male | Pseudo-scent | Green dog/Working dog |
| Krilling | 1.5 | Belgian Malinois | Male | Sweat/Saliva /Pseudo-scent | Dog trained by OBI-UNISON project (Sweat) |
| Leia | 1.5 | Golden Retriever | Female | Sweat/Saliva /Pseudo-scent | Epilepsy dog/ Dog trained by OBI-UNISON project (Sweat and saliva) |
| Marco | 2 | German Braco | Male | Pseudo-scent | Dog trained by Bio Detection K9 (USA) |
| Dominique | 2 | Belgian Malinois | Male | Pseudo-scent | Dog trained by Bio Detection K9 (USA) |
| Sam | 3 | German Shepherd | Male | Sweat/Saliva /Pseudo-scent | Dog trained by OBI-UNISON project (Sweat) |
| Orion | 3 | Labrador Retriever | Male | Saliva | Rescued dog |
| Mike | 3 | German Shepherd | Male | Sweat/Saliva /Pseudo-scent | Dog trained by OBI-UNISON project (Sweat) |

BDK9, this pseudo-scent closely resembles the odor associated with SARS-CoV-2. BDK9 has employed this pseudo-scent to train dogs to detect SARS-CoV-2 and screen individuals at large gatherings. The positive controls consisted of sweat samples obtained from patients diagnosed with SARS-CoV-2, the negative controls consisted of sweat samples from individuals confirmed to be negative for SARS-CoV-2, all samples confirmed through RT-PCR testing. In the case of positive samples obtained from the Progreso Health Center and the Biostudio Private Laboratory, these had already been previously diagnosed with an antigen test or RT-PCR, respectively. However, all samples (positive and negative) were subsequently tested and confirmed in this study by RT-PCR for consistency and standardization.

## Sampling

Sweat samples from SARS-CoV-2 positive and negative humans were used for training evaluation of the dog. All positive samples (n = 90) were obtained from two sources, the Progreso Health Center in Progreso, Yucatán, and the private laboratory Biostudio from Merida, Yucatán. The negative samples (n = 289) were obtained from people of Merida City. The sampling process included a briefing of the objectives of the research, obtaining an informed consent, application of an epidemiological questionnaire, and obtention of the sweat samples. Briefly, before sampling both negative and positive people, each was asked for her/his willingness to participate in the research, and in the case of a positive answer, a briefing on the objectives of the project and the subsequent use of the samples and epidemiological information was provided. The patient was then provided an informed consent form to read and sign, accepting her/his participation in the project (S1 File). An epidemiological questionnaire was used to obtain data on symptoms and medical history of the patient. The questionnaire collected information on full name, age, gender, diagnosed chronic diseases, headache, diarrhea, fever, loss of taste, loss of smell, cough, runny nose, sore throat, body ache, chest pain, nausea, days with symptoms, treatment and days on medication (if provided), and contact with confirmed SARS-CoV-2-positive people. These data are available under request to Juan Manuel Mancilla-Tapia. All samples were obtained in Mérida; Yucatán between September and November 2021.

All positive samples were colected from individuals presenting symptoms such as a cold, fever, headache, and/or diarrhea, who attended either the Progreso Health Center in Progreso, Yucatán or the private Biostudio laboratory asking for a test for SARS-CoV-2. Two samples were obtained from each person at the Progreso Health Center by an expert technician. The first sample was a swab from the throat and nasopharynx of all patients following standard procedures recommended by the Mexican health authority for the RT-qPCR performed by the Public Health Laboratory of Yucatan [19, 20]. The second swab was also obtained from the throat and nasopharynx of the same patient by the same technician, and it was immersed immediately in a 10 ml Falcon tube with DNA/RNA shield fixative (Zymo Research®, Irving CA) and posteriorly used for the RT-qPCR following the procedures suggested by CDC and Pearson and colleagues [16, 21]. The other source of positive samples was acquired through the private laboratory Biostudio (https://www.biostudio.com.mx/). People attending to the laboratory for an antigen or RT-qPCR test for SARS-CoV-2 and tested positive were asked whether they would be willing to provide a swab from the throat and nasopharynx. The swabs obtained were immersed immediately in a 10 ml Falcon tube with RNA shield fixative (Zymo Research®; Irving CA) and subsequently used for RT-qPCR. The inclusion criteria for SARS-CoV-2 positive patients were (1) age range between 14 (with parents' consent) and 60 years old, (2) ≤ 9 days of symptoms, and (3) positive for SARS-CoV-2 confirmed by RT-PCR or by an antigen test. All negative samples were swabs from the throat and nasopharynx of healthy

people from the research center (CINVESTAV-IPN Mérida), from football and baseball teams, or from relatives and acquaintances in Mérida, Yucatán. The swabs of these people were individually fixed in a 10 ml Falcon tube with RNA shield fixative (Zymo Research® Irving CA) and used for RT-qPCR with the Norgen Biotek Corp kit. During the field work at the Progreso Health center, Biostudio laboratory or the facilities at which negative people were sampled, adequate personal protective equipment was always used while collecting the throat and nasopharynx samples. All sweat samples, regardless of whether the tests indicated they were positive or negative for SARS-CoV-2, were handled by following the safety measures (e.g. KN 95 face masks, nitrile gloves) recommended by the Mexican health authorities [19]. During sample collection, the patients took their own sweat samples under technical supervision. To collect the sweat samples, each patient was given a pair of non-sterile, dust-free nitrile gloves and a resealable Ziploc® bag containing two 7 cm high, 4 cm diameter amber glass flasks with plastic (for dogs sniffing) or bakelite (for chromatography) caps sterilized in an autoclave, and six pieces of new, sterile Jaloma® odorless gauze (10 cm × 10 cm). The patient was asked to rub her/his neck, face, and forearms for 1 minute with two gauze pads, and to insert them into one of the flasks. Subsequently, the patient was asked to place two gauze pads under each armpit for 10 minutes. After this time, the patient was instructed to insert the gauze pad with sweat samples into the glass flask, to close it, and to place it back in the resealable bag. This process was approved by the General Hospital Investigation Committee and the Bioethics and Safety Committee of the University of Sonora. No fixatives were added to the sweat samples for training of the dogs. The samples for molecular biology and sweat were then transported in separated coolers to the laboratory and kept there at 4°C until they were processed.

## Molecular biology

Nasopharyngeal samples that were preserved in DNA/RNA Shield fixative were used for the extraction of genomic RNA. Viral RNA extraction was performed in a biosafety cabinet class II, type A2 (Labconco, Kansas, USA) using the Quick-RNA Viral Kit 200 extraction kit preps R1035 (Zymo Research; Irving CA) following the manufacturer's instructions. Appropriate personal protective equipment was worn while samples were processed. N95 masks, nitrile gloves and all personal protective equipment recommended by the Mexican health authorities [19]. This process was approved by the General Hospital Investigation Committee and the Bioethics and Safety Committee of the University of Sonora. Each individual sample was processed using the One-step Real-Time Reverse Transcriptase Polymerase Chain Reaction (RT-qPCR) method according to the manufacturer´s instructions in concordance with the recommendations of the Mexican health authorities [20]. The kit of probes and primers used was manufactured by Norgen Biotek Corp (Ontario, Canada). This kit uses two regions (N1 and N2) of the nucleocapsid N gen of the SARS-CoV-2 virus and a human control gene (RP). The enzyme used was GoTaq® Probe 1-Step RT-qPCR System, A6121 (Promega Corporation, Wisconsin, USA). All nasopharyngeal samples were processed using a CFX Opus 96 Real-Time PCR thermal cycler System (Bio-Rad Laboratories, Inc., California, USA). All RNA extraction procedures, reverse transcription of RNA into complementary DNA (cDNA) with transcriptase inverse, RT-qPCR and quantification of the amount of target nucleic acids, were carried out at the Biostudio clinical analysis laboratory, Mérida, Yucatán by personnel from CINVESTAV-IPN Mérida (DHM). For each sample, the cycle threshold (Ct) was recorded. Ct values were used to indirectly quantify the relative amount of viral RNA in a sample [8, 22]. The obtention of the RT-qPCR results lasted at least two days. Thus, all the samples exposed to dogs were at least two days old.

## Statistical analysis of the epidemiological data

The epidemiological data of the patients were used as independent variables, and the number of dogs detections (see below) and the Ct values for the N1 and N2 regions were used as dependent variables in a canonical correlation analysis. Both independent and dependent variables were properly standardized using Z.transformation and CANOCO 5 software was used (ter Braak &Smilauer, 2022) for the analysis.

## Calculation of viral load

To calculate the SARS-CoV-2 viral load using the Ct values, a standard curve was produced for the N1 and N2 regions using the positive control 2019-nCoV PC67102 (Norgen Biotek Corp, Ontario, Canada), which had a known concentration of 200,000 copies per microliter (copies/μL). Dilutions of this control were made to identify a range of the limit of detection (LoD) of the positive control by the RT-qPCR. Six dilutions were used trying to cover the whole range of concentrations of viral copies/μL. Ct values resulting from the positive control dilutions were then used to construct a curve. The positive control dilutions were transformed to logarithms base 10 and plotted against the Ct values obtained for those dilutions by RT-qPCR. A linear regression was applied afterwards using the logarithmic values of the positive control dilutions as the independent variable and the Ct values associated to those dilutions as the dependent variable. The least-squares regression method was applied, and the values of the slope and y-intercept calculated. Once the linear regression equation with the slope and y-intercept was established, the number of viral copies of each patient could be estimated based on their respective Ct values. Calculation of the relative number of viral copies per microliter for each patient was performed for both N1 and N2. All these calculations were made using InfoStat software [23].

## Exposure of the dogs to the sweat samples

For each trial, what was referred to as "an odor line" was used This stainless-steel odor line had four holes. In three of the four holes, three stainless-steel saltshakers were allocated, each with two gauze pads with sweat of a negative person confirmed by RT-qPCR, and in the fourth hole one saltshaker was placed containing a positive sample also corroborated by RT-qPCR. Each dog was passed through all the positive and negative samples in the line for definitive detection in the same line. In this way, the line-up used in the present study was essentially the same methodology proposed by Grandjean and colleagues [1]. Thus, for each trial in a line-up, the dog sniffed each of the four saltshakers and the dog handler asked the dog to look for the positive sample. The saltshaker the dog marked was considered the definitive identification decision for that trial. For each trial, new fresh positive and negative samples were always used, and none of the previous samples were used again. The positive and negative samples were allocated randomly by the data recorder, and neither the dog handler nor the dog knew where the positive samples were, because both were asked to look in a different direction when the saltshakers were allocated in the line-up (double-blind strategy). The data recorder indicated that the line-up was ready, and it was at this moment that both the dog handler and the dog faced the line-up. Once the dog sniffed all the saltshakers and marked one (by sitting or lying on it) the dog handler made a signal (upright closed fist) to indicate that the trial had finished. The data recorder indicated verbally whether the mark was correct, and if so, the dog handler immediately rewarded the dog with a toy (Kong), allowing him/her to play two–three minutes. The testing period lasted for 12 weeks, during which time none of the dogs showed disease signs.

## Sensitivity and specificity

Sensitivity (the ability of the test to correctly detect ill patients out of those who do have the condition) and specificity (the ability of the test to correctly reject healthy patients without a condition) were calculated as recommended by Johnen and colleagues [24], and their 95% confidence intervals (CI) were calculated with the Clopper–Pearson's method, using the package epiR [25]. It was considered that if a 95% CI does not overlap 50%, which is the randomness region, that 95% CI could be accepted significantly different from a random choice. The procedures recommended by Trevethan were followed to calculate sensitivity and specificity [26]. The minimum number of samples to be sniffed for adequate study power was calculated assuming a 15% of prevalence of SARS-CoV-2 in the population. This was a rather conservative estimation of prevalence since by November 2021, being in the middle of the third wave of SARS-CoV-2 at Mérida, with the Delta variant as the dominant one. The probability of type 1 error, determining that there is a difference when such difference does not actually exist, was established at 0.05. The power of the analysis to detect a difference between groups when such a difference exists was assumed to be 80%. All the calculations for the number of samples to be sniffed were made with ClinCalc.com (https://clincalc.com/stats/samplesize.aspx). These calculations gave us a sample size of 94 to have an adequate study power.

## Sum of dog's detection

The information of a dog's detections after exposure to sweat samples from positive and negative people to SARS-CoV-2 was compiled (S2 Table) together with the Ct values of the RT-qPCR test of each person participating in the experiment. With this information, it was possible to know how many dogs and what dogs detected correctly (or not) a positive or negative sample previously corroborated by RT-qPCR. The samples were then ordered based on the number of correct detections made by the dogs. For example, a positive sample could have been correctly detected by 10 out of 11 dogs. Thus, all these detections were summed, and this variable were called "sum of dogs' detections" for that specific sample. The same process was applied to those samples correctly identified by nine dogs, eight dogs, seven dogs up to those detected correctly by one dog. An Excel file was created containing both the sum of dogs' detections for positive and negative samples, as well as the two Ct values associated with each sample.

The best-fitting model was determined describing the statistical association between "sum of dogs' detections" as a dependent variable and the Ct values as an independent variable. The relationship between these two variables was analyzed using the following models: linear, power, polynomial 2$^{nd}$ degree, logarithmic and gaussian. All models were fitted using *nls* function (Nonlinear Least Squares estimation) of R programming language [27]. The best model was chosen through a multi-model approach based on Information Theory [28], positioning the most parsimonious models based on the lowest Akaike's Information Criterion value (AIC, Akaike, 1983), calculating Akaike's differences ($\Delta i$) and Akaike's weights (wi).

## Results

A total of 379 samples were obtained of both sweat and nasopharyngeal swabs for RT-qPCR from people from Merida, Yucatán between September and November 2021. The positive sweat samples were obtained from a total of 90 people from two sources, the Progreso Health Center in Progreso, Yucatán, and the private laboratory BioStudio at Merida, Yucatán. The negative samples (n = 289) were obtained from people of Merida City. The whole sample comprised 174 (46%) women and 205 (54%) men. The group of positive people comprised 60 (67%) women and 30 (33%) men, and the group of negative people was 119 (41%) women and

170 (59%) men. There were significant differences between the SARS-CoV-2 positive and negative groups in the proportion of women and men (Fisher's exact test, difference between proportions = -0.24, p < 0.0001). In the group of positive people, there was a significant difference in the proportion of women and men (Fisher's exact test, difference between proportions = -0.27, p< 0.001). In the group of negative people there was also a significant difference in the proportion of women and men (Fisher's exact test, difference between proportions = -0.20, p< 0.0001).

The age range of the whole sample (positives and negatives) was between 14 and 80 years, with $37 \pm 14$ years for women and $38 \pm 15$ years for men. The age in the positive group was $38 \pm 14$ years for women and $39 \pm 16$ years for men; there was no significant difference in age between both sexes (Student's $t_{0.05}$ = -0.16, p = 0.87). The age in the negative group was $37 \pm 14$ years for women and $38 \pm 15$ years for men; there was no significant difference in age between both sexes (Student's $t_{0.05}$ = -0.92, p = 0.36). There were no differences in the mean age between the SARS -CoV-2 positive and negative groups (Student's $t_{0.05}$ = -0.56, p = 0.57).

## Sensitivity and specificity

Table 2 shows the sensitivity and specificity for the 11 second generation dogs exposed to sweat samples of people from Merida compared with the results of RT-qPCR. Five of 11 dogs had a marginal performance since their 95% CI overlapped the randomness region (50%). In contrast, Marco, Mike, Sam and Leia had sensitivity values between 67% and 69%, and their 95% CI did not overlap the randomness region. The dogs with the best performance in sensitivity were Krilling and Orion with 72% and 87%, and their 95% CI did not overlap the randomness region. In the case of Rex, he had a sensitivity of 77%, but his 95% CI overlapped the randomness region. With respect to specificity, with the exception of Mike, Mona and Sam that had values between 87% and 89%, all other dogs had values of 90% specificity or above. Rex had the best value, with 97% specificity, and in no case did the 95% CI of the dogs overlap the randomness region.

## Statistical analysis of the epidemiological data

There was a significant statistical association between the epidemiological data of the patients, and the number of dogs detections and the Ct values in canonical correlation analysis for both

**Table 2. Sensitivity and specificity of 11 dogs trained to detect SARS -CoV-2 from the sweat of positive and negative people from Mérida, Yucatán compared with the reference test (RT-qPCR).**

| Name | n | Sensitivity (%) | 95% CI | Specificity (%) | 95% CI | PPV (%) | 95% CI | NPV (%) | 95% CI |
|---|---|---|---|---|---|---|---|---|---|
| Dominique | 410 | 61 | 49–71 | 93 | 90–95 | 69 | 57–79 | 90 | 86–93 |
| Krilling | 410 | 72 | 61–81 | 91 | 87–94 | 68 | 58–78 | 92 | 89–95 |
| Leia | 352 | 69 | 59–78 | 90 | 86–94 | 74 | 64–83 | 88 | 83–92 |
| Lucky | 194 | 68 | 48–84 | 92 | 87–96 | 59 | 41–76 | 94 | 90–97 |
| Marco | 412 | 67 | 56–77 | 92 | 88–94 | 68 | 57–78 | 91 | 88–94 |
| Mike | 408 | 67 | 56–77 | 87 | 83–90 | 58 | 48–68 | 91 | 87–94 |
| Mona | 414 | 53 | 42–64 | 88 | 84–92 | 55 | 44–66 | 88 | 83–91 |
| Sam | 410 | 68 | 57–78 | 89 | 85–92 | 63 | 53–73 | 91 | 87–94 |
| Orion | 96 | 87 | 60–98 | 91 | 83–96 | 65 | 41–85 | 97 | 91–100 |
| Rex | 93 | 77 | 46–95 | 97 | 91–100 | 83 | 52–98 | 96 | 90–100 |
| Ziggy | 74 | 58 | 33–80 | 91 | 80–97 | 69 | 41–89 | 86 | 75–94 |

CI, 95% confidence interval; n, number of trials. In this case the number of trials is larger than the number of samples because in many cases the same positive sample was positioned in a different hole in the line to be sniffed by the dog; NPV, negative predictive value; PPV, positive predictive value; RT-qPCR, Real-Time Reverse Transcriptase Polymerase Chain Reaction.

the first (F = 0.1; p = 0.0002, n = 367) and all canonical axes (F = 1.7; p = 0.0002). However, the percentage of explained variance of the analysis was only 6.03%, which was considered irrelevant. Thus, no further consideration was made about this analysis.

## Calculation of Ct and viral load

The minimum detection limit of the SARS-CoV-2 kit used in this project was 5 copies/μL, which is equivalent to an approximate Ct of 38.68 and 38.95 for the N1 and N2 regions respectively. On the other hand, detection minimum and optimum were 10 copies/μL, that is, a Ct of 32.3 and 33.22 for N1 and N2 respectively. The estimation of the number of copies among the positive patients was variable, since the minimum value of viral copies per μL found in a patient was 4.53–8.40 (6.46 ± 2.74) belonging to a Ct of 37.89–35.70 of the N1 and N2 regions respectively. The positive patient with the highest number of viral copies per μL was 180,003.65–183,164.19 (181,583.92 ± 2234.84) with a Ct of 13.63–13.15 of the N1 and N2 genes respectively. The relationship between the Ct values of the N1 region obtained from people positive to SARS-CoV-2 and their viral load was a negative and significant linear relationship (Y = -3437.62(X) + 122891.96; $F_{1,9}$ = 100.71; N = 10; $P < 0.0001$; $R^2$ = 0.93) in which, the higher the values of Ct, the lower the viral load. The same kind of relationship applied for the N2 region (Y = -4214.12(X) + 150653.61; $F_{1,9}$ = 24.98; N = 10; $P < 0.01$; $R^2$ = 0.83).

## Statistical association between Ct values, viral load and sum of dog's detection

Table 3 shows the Ct values at which a certain number of dogs out of 11 was able to detect SARS-CoV-2. For the N1 region, between 63 and 91% of the dogs detected SARS-CoV-2 optimally when Ct range is between 18.49 and 29.17 (23.83 ± 5.34). In contrast, when the Ct range falls below 17.92 or is above 33.00 (25.46 ± 7.54) only 9 to 27% of the dogs were able to detect the virus. In the case of the N2 region between 63.63 and 90.90% of the dogs were able to detect SARS-CoV-2 optimally when Ct range was between 24.07 and 26.69 (25.38 ± 1.31), clearly a very narrow range. In N2, for Ct values lower than 20.13 or higher than 39.03 (29.58 ± 9.45), the dogs were unable to detect the virus. The dog's detections followed a gaussian model (S1 Table). Indeed, the Akaike´s information criterion was lower for the gaussian model, compared with values for other statistical models (S1 Table). This suggest that for Ct values between 0 and 15, there was a low detection capacity for both N1 and N2 regions (Fig 1A and 1B). However, for Ct ranges between 16 and 29 the detection capacity was high, and from Ct values of 30 and higher the detection capacity diminished rapidly for both N1 and N2 regions (Fig 1C). In fact, the coefficient of determination of this model was $R^2$ = 0.52, with mean Ct

**Table 3. Cycle threshold values at which a specific number of dogs out of 11 dogs was able to detect SARS-CoV-2 in samples from Merida, Yucatán, México.**

|  | Number of dogs detecting SARS-CoV-2 out of 11 dogs | Mean value of the cycle threshold (Ct) for the N1 fragment | Mean value of the cycle threshold (Ct) for the N2 fragment |
|---|---|---|---|
| Positives | 7 to 10 | 23.83 ± 5.34 | 25.38 ± 1.31 |
|  | 4 to 6 | 24.59 ± 7.00 | 26.15 ± 7.88 |
|  | 1 to 3 | 25.46 ± 7.54 | 26.94 ± 8.57 |
|  | 0 | 27.78 ± 8.00 | 29.58 ± 9.45 |
| Negatives | 10 | 38.04 ± 2.75 | 40.59 ± 3.19 |
|  | 1 to 2 | 38.41 ± 2.54 | 39.34 ± 3.42 |
|  | 3 to 4 | 40.54 ± 4.46 | 40.33 ± 2.57 |

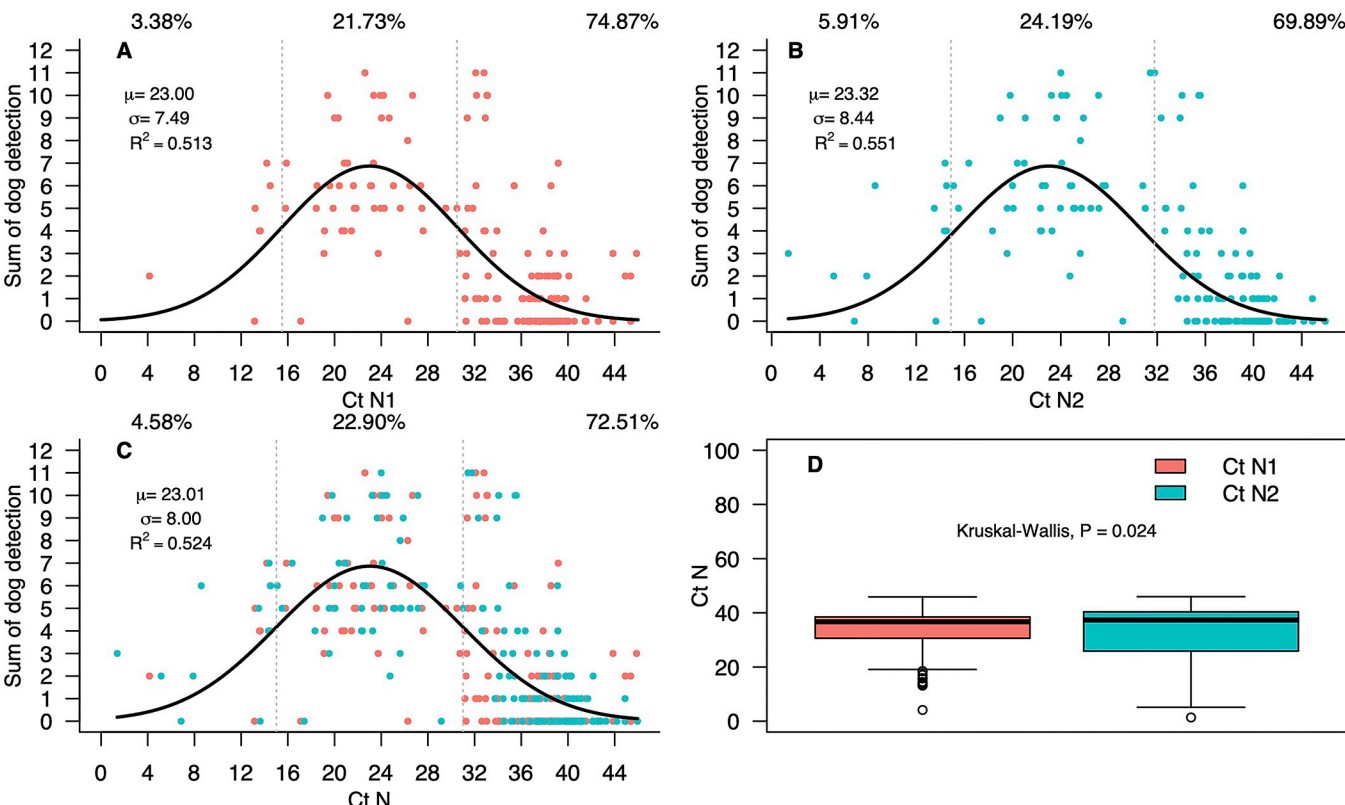

**Fig 1. Gaussian model fitted of the number of dogs detections of SARS-CoV-2, depending on the cycle threshold (Ct) values of RT-qPCR tests for SARS-CoV-2.** A) Gaussian model of dog's detections with respect the Ct values of the N1 region of the nucleocapsid gen for the SARS-CoV-2 virus. B) Gaussian model of dog's detections with respect the Ct values of the N2 region of the nucleocapsid gen of the SARS-CoV-2 virus. C) Overlapped gaussian model of dog's detections with respect the Ct values of the N1 and N2 regions of the nucleocapsid gen of the SARS-CoV-2 virus. D) Kruskal-Wallis test of the frequency distribution values of the dogs detections for the N1 and N2 regions of the nucleocapsid gen of the SARS-CoV-2 virus.

value of 23.04 ± 8.00. There was a significant difference in the dispersion of the Ct values for both N1 and N2, with values of the former behaving closer to the gaussian model (Fig 1D).

## Discussion

The first hypothesis was that second generation of trained dogs would have better levels of sensitivity and specificity compared to the previous generation [7]. However, this hypothesis was only partially upheld, only 45% dogs achieved sensitivity values surpassing those considered random (> 50% sensitivity). There was an improvement in the detection ability of the second-generation dogs with a sensitivity range between 58% and 80% [7]. This suggests a potential heritability of the trait. Regarding specificity, there was also an improvement with values for this second generation [7]. When comparing the sensitivity and specificity results of the dogs with those of the Rapid Antigen Detection immunoassays (RADs) available in the market, they fall within the range of values of these commercial tests. In fact, in several published studies, the dogs' results exceeded those of the RADs when both diagnostic alternatives were used in Points of Care (POC). A notable contrast was observed in Lambert-Niclot et al. [29], where the dogs exhibited sensitivity values between 67 and 87%, while the COVID-19 Ag Respi-Strip (Coris Bio-Concept, Gembloux, Belgium) in the aforementioned study displayed a sensitivity of 50% compared RT-qPCR values. Another contrasting case was observed in the study by Thommes et al. where the Novel Coronavirus (2019-nCov) Antigen Detection Kit

(CLMSRDL, Sichuan Mass Spectrometry Biotechnology Co., Ltd, Chengdu, Sichuan) exhibited only a 60% sensitivity compared to RT-qPCR results [30]. RADs such as PanbioTM COVID-19 Ag Rapid test (Abbott, Chicago, Illinois), DIAQUICK COVID-19 Ag Cassette (DIALAB, Wiener Neudorf, Austria), and SARS-CoV-2 Rapid Antigen Test (Roche Diagnostics Deutschland GmbH, Mannheim, Germany) presented excellent sensitivity results, with 83.3% (95%CI: 58.6–96.4%; N = 18 (15/3), 100% (95%CI: 73.5–100%; N = 12 (12/0)) and 100% (66.4–100%; N = 9 (9/0)) respectively [30]. Another study that corroborated the sensitivity of the PanbioTM COVID-19 Ag Rapid test was conducted by Albert et al. [31] who found a sensitivity of 79.6% (95% CI: 67.0–88.8–100%; N = 412) [31]. Finally, in an extensive study on diagnostic accuracy of RADs, Dinnes et al. [32] in a Cochrane systematic review, found that RADs sensitivities varied from 34.1% (95%CI: 29.7–38.8%; Coris Bioconcept test) to 88.1% (95%CI: 84.2–91.1%; SD Biosensor Standard Q) in symptomatic people [32]. Thus, the results obtained by the dogs in this study fall within this range and can be considered an important tool for discriminating between positive and negative people with SARS-CoV-2.

In terms of specificity, the dogs also performed well. However, comparisons here are more challenging, especially because most of the studies on people were conducted in hospitals or POC settings and primarily on symptomatic individuals. Albert et al. [31] found a specificity value of 100% (95% CI: 98.7–100%; N = 412) for the PanbioTM COVID-19 Ag Rapid test [31]. In their systematic review, Dinnes et al. [32] found that the average specificities of symptomatic and asymptomatic people for the 16 antigens test they studied was 99.6% (95% CI: 99.0–99.8%). In practice, it means that the dogs could potentially confuse other respiratory infections for COVID-19, but they could still effectively distinguish infected from uninfected individuals. However, the most effective way to increase the dogs' specificity in detecting SARS-CoV-2 is by implementing the method proposed by ten Hagen et al. [5], which involves exposing them to various viral respiratory tract infections. These authors achieved high specificity levels using this method. While the dogs' specificities do not yet meet the WHO's (2021) standards ($\geq$97–100%), their ability to perform rapid detections on a large scale remains a valuable aspect. However, it is necessary recognize that for dogs to be considered a viable complementary tool, their success rate needs to be further analyzed in different scenarios, considering the prevalence of the disease in the population. Although dogs may not be a direct substitute for molecular testing in terms of specificity, their strategic use in specific contexts, could have a positive impact on outbreak containment, provided that limitations are properly considered.

The second hypothesis suggested that dogs would perform significantly better in detecting SARS-CoV-2 within a specific range of Ct values, with substantial variability in detection capacity outside this range, potentially resulting in some dogs outperforming others. The hypothesis was confirmed, as all dogs capable of detecting the virus did so within the specified Ct range. A gaussian frequency distribution was observed, indicating that dogs were unable to reliably detect the virus below Ct values of 17.92 and above 33.00. The optimal range for detecting SARS-CoV-2 in both the N1 and N2 regions was found to be between Ct values of 18.49 and 29.17. These performance values align closely with those observed in RADs detecting SARS-CoV-2 in the literature. For example, Albert et al. [31] found that the PanbioTM COVID-19 Ag Rapid test performed well when Ct values were below 25 [31]. However, these authors do not provide the lower limit at which the test does not detect the infection. Thommes et al. [30] found that, for Ct values lower than or equal to 25, the performances of PanbioTM COVID-19 Ag Rapid test, DIAQUICK COVID-19 Ag Cassette (DIALAB, Wiener Neudorf, Austria) and SARS-CoV-2 Rapid Antigen Test (Roche Diagnostics Deutschland GmbH, Mannheim, Germany) were as good as described in the sensitivity section above. However, when the Ct values are higher than 30, the sensitivity of these three RADs decreases

dramatically to values between 25.6% to 46%. Finally, Dimmes et al. [32] found that the 16 diagnostic tests they studied performed very well when Ct values were below 25 and people had symptoms within the first week of infection. Thus, the dogs performed comparably to standard RADs, reinforcing their potential as an important tool for discriminating between positive and negative cases of SARS-CoV-2 in real life settings.

There is no doubt that Mexican dogs can improve their sensitivity and specificity values by applying methodologies such as those suggested by ten Hagen et al. [5] exposing them to different viral respiratory tract infections. Additionally, further training program, as suggested by ten Hagen et al. [5], would be useful to improve their performance in large-scale activities. Recently, such an additional training program has been proposed by Meller et al. [4] as an effective way to keep the dogs prepared for real-life scenarios [4].

An anecdotal experience developed by one of us (JMT), involved a contract with the Purina Pet Care company (Nestle ™) in Mexico, where employees from three pet food factories were screened daily during three months, for a total of 14,000 workers. The result of this experience revealed no confirmed cases of SARS-CoV-2 detected by the dogs during this period at any of the factories.

Additionally, the low specificity of the Mexican dogs reported in the contract mentioned above led to the implementation of preventive measures to mitigate the introduction of other respiratory tract infections in the factories. A summary of the contract with Purina Pet Care can be found at https://purina.com.mx/purina/purina-sociedad/obi-k19. Although it was not feasible to conduct the required number of RT-PCR tests to confirm the negative status of all factory workers, this, coupled with the results obtained by ten Hagen et al. [4] at concerts, demonstrating the utility of dogs in screening large numbers of people.

It is important to note that the WHO has requested that RAD should demonstrate sensitivity values ≥80%, and specificity values ≥97–100% [33]. In real life these are extremely stringent requirements since the human populations are composed of both symptomatic and asymptomatic individuals and it is extremely difficult to reach and maintain such sensitivity and specificity values through time. A much more practical view is the one proposed by Larremore et al. [34] who suggest that more important than sensitivity is the frequency at which a diagnostic test can be applied [34]. In fact, these authors suggest that effective screening of SARS-CoV-2 depends on the frequency of testing and the speed of reporting rather than high test sensitivity. In this regard, dogs possess a significant advantage, because they can screen hundreds of people in short time, compared with the more expensive and logistically complex application of RADs.

Finally, while our study primarily focused on the detection of SARS-CoV-2, it is important to acknowledge the broader context of emerging viral diseases, particularly those caused by RNA viruses. The high mutation rates of RNA viruses pose a continuous threat, as evidenced by recent pandemics like COVID-19 [35]. RNA viruses such as Zika, Ebola, and Influenza have demonstrated their potential to cause widespread outbreaks [35, 36]. The ability to rapidly detect such viruses is critical in managing future pandemics. Although the dogs in this study did not meet the specificity standards set by the WHO, their capacity to perform large-scale screening in short periods of time makes them a valuable tool in the fight against emerging RNA viral diseases, especially in areas where testing resources are limited or expensive. Expanding the application of detection dogs to other RNA viruses could enhance global preparedness for future outbreaks.

In conclusion, the ability of the dogs to detect the virus is related to the threshold cycle (Ct) values of the RT-qPCR test in a specific range, in which the dogs demonstrated a better detection capacity. This relationship suggests that dogs may be effective in discriminating between positive and negative cases of SARS-CoV-2 in real-life settings, especially in resource-limited

settings. Likewise, the results of this study show that, although there was an improvement in the detection capacity of the second generation of dogs trained to detect SARS-CoV-2, there are still challenges in terms of sensitivity and specificity. Although six of the eleven dogs demonstrated a sensitivity between 67% and 87%, and all dogs showed a specificity between 87% and 92%, some did not reach sensitivity values considered optimal. However, compared to commercially available rapid antigen detection tests, in some cases, the dogs' results exceeded those of such tests. Although the dogs did not meet the sensitivity and specificity standards set by the WHO, their ability to perform large-scale screening in short periods of time makes them a valuable tool in the fight against the spread of COVID-19, especially in areas where RT-qPCR and rapid antigen testing are limited or expensive. However, the importance of continuing research and training to further improve the ability of dogs to detect SARS-CoV-2 and other viral respiratory infections is noted.

## Supporting information

**S1 File. Informed consent format.**
(DOCX)

**S1 Table. Comparison of the fit linear, power, polynomial (2° degree), logarithmic and gaussian models to the dog's detection data of S2 Table.** The best model was that with the lowest value of the Akaike´s information criterion.
(DOCX)

**S2 Table. Results of a dog's detections after exposure to sweat samples from positive and negative people to SARS-CoV-2, together with the Ct values of the RT-qPCR test of each person participating in the experiment.**
(XLSX)

## Acknowledgments

The authors were indebted to Karla Valenzuela Lozano (nurse and sample collection) and Georgina Villegas, Enrique Claussen (former secretary of Health of the state of Sonora). The authors thank Clara Vivas-Rodríguez, Gregory Arjona-Torres, Francisco Puc-Itza and Oscar Arturo Centeno-Chalé for support with the field and laboratory work.

## Author Contributions

**Conceptualization:** Víctor M. Vidal-Martínez.

**Data curation:** Lilia C. Soler-Jiménez, David Hernández-Mena.

**Formal analysis:** Víctor M. Vidal-Martínez, Iván Velázquez-Abunader.

**Funding acquisition:** Juan Manuel Mancilla-Tapia.

**Investigation:** Víctor M. Vidal-Martínez.

**Methodology:** Juan Manuel Mancilla-Tapia, Lilia C. Soler-Jiménez, Iván Velázquez-Abunader, David Hernández-Mena.

**Project administration:** Juan Manuel Mancilla-Tapia.

**Writing – original draft:** Víctor M. Vidal-Martínez.

**Writing – review & editing:** Juan Manuel Mancilla-Tapia, Lilia C. Soler-Jiménez, Iván Velázquez-Abunader, Matilde Jiménez-Coello, Antonio Ortega-Pacheco, David Hernández-Mena.

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
