## [Decision Letter · Decision Letter 0]

19 Feb 2024

PONE-D-23-33291At what cycle threshold level are dogs able to detect SARS-CoV-2 in humans?PLOS ONE

Dear Dr. Soler-Jimenez,

Thank you for submitting your manuscript to PLOS ONE. After careful consideration, we feel that it has merit but does not fully meet PLOS ONE’s publication criteria as it currently stands. Therefore, we invite you to submit a revised version of the manuscript that addresses the points raised during the review process.

We look forward to receiving your revised manuscript.

Kind regards,

Julian Ruiz-Saenz

Academic Editor

PLOS ONE

Journal Requirements:

3. Thank you for stating the following financial disclosure: "JMMT Funding acquisition, project supported by the Mexican Government

(CONAHCyT) for training dogs in the bio-detection of SARS-CoV-2 at the OBI-K19 training center. Agreement Number: 000000000317533 Consejo Nacional de Humanidades, Ciencias y Tecnologías https://conahcyt.mx/"

Reviewers' comments:

Reviewer's Responses to Questions

**Comments to the Author**

1. Is the manuscript technically sound, and do the data support the conclusions?

Reviewer #1: Yes

Reviewer #2: Partly

2. Has the statistical analysis been performed appropriately and rigorously? 

Reviewer #1: Yes

Reviewer #2: I Don't Know

3. Have the authors made all data underlying the findings in their manuscript fully available?

Reviewer #1: Yes

Reviewer #2: No

4. Is the manuscript presented in an intelligible fashion and written in standard English?

Reviewer #1: Yes

Reviewer #2: No

5. Review Comments to the Author

Reviewer #1: Comments to the Author

In the manuscript entitled “At shat cycle threshold level are dogs able to detect SARS-CoV-2 in humans?”, the authors present the findings of high detection capacity of COVID-19 by dogs for Ct ranges between 16 and 29.

They conclude the possibility and utility for the rapid detection way of COVID-19 by trained animals in case of screening the crowd. This study provides valuable information, additional explanations should be required to improve the manuscript for the readers.

Major points

1. How many consecutive judgements by the dogs maintain the sensitivity and specificity? In the discussion section, the authors described “they (the dogs) can screen hundreds of people in a few hours”. but the testing period lasted for 12 weeks in this study. The discussion might be overestimated.

2. The mechanism of detecting the SARS-CoV-2 by the dogs smelling the sweats seemed not to be clear. Did the severity of COVID-19, the patients’ immune status or the kinds of the variants of SARS-CoV-2 influence? If the possibility was evaluated, the data should be added.

3. Did the Ct values of sweats and nasopharyngeal samples in one person show the same?

Minor points:

1. In figure 1D, the range of Ct values between 60 and 100 might not be needed.

2. Figure legends found in the results section. It should be moved to the appropriate section.

3. How often the training is required to maintain the ability of detecting the virus?

Reviewer #2: The authors performed the COVID-19 detection by trained dogs by using a gauze, and cycle threshold by RT-PCR against N1 and N2 SARS-CoV-2 gene. Some aspects should be addressed by the authors.

1. The abstract should be improved and then to impact the data obtained.

2. A large introduction is presented, and much information are outside for this.

3. There is a lack of information about the training procedure of the dogs. How was performed, time or period of training, samples used, pseudo-scent, what is the pseudo-scent, positive controls. Negative controls, confuse controls, etc.

4. Several repetitive information is readily available, this is confused to read.

5. The determination of copies is for sample or individual.

6. To many data in the manuscript, could be better a table to observe it.

7. Why the authors used number of virus copies, there is lack of information about the virus DNA detected by dogs. Do you have background about DNA detected by dogs?. A logic thought is the dog detects organic compounds that inert compounds (DNA). In this line, high viral load is directly related to high viral protein, is this correct?

8. The sensitivity and specificity values by each dog are different, obviously, different data should be obtained.

9. The data craving to compare rapid antigen detection with the dogs than the DNA detection. A real situation is, which is better detection for COVID-19 disease, dogs or rapid detection strip?

10. Regarding the clinical data of the patients included in the protocol, they were severe, mild COVID-19? This could discriminate among different clinical diagnoses.

11. The dogs are not machine which have loaded a program, they should be trained previously and then to challenge with the target. However, any dog doesn’t have the characteristic to work for specific work.

12. An important question is how much viral (antigen) protein is secreted (outside of the cell) and be detected by the dog. Of course, if you intended to work with dogs. Another is, if high viral load is directly proportional to high secreted proteins.

13. How reinforced for dogs were performed. The time involved to read all the samples. The read performed were in close room or in an open site.

14. Once obtained all the samples were immediately reads or were stored.

15. There is a lack of a conclusion.

16. The abstract should be improved and then to impact the data obtained.

17. The title of the manuscript is not related to data presented.

6. PLOS authors have the option to publish the peer review history of their article (what does this mean?). If published, this will include your full peer review and any attached files.

Reviewer #1: No

Reviewer #2: **Yes: **MAURICIO SALCEDO

---

## [Author Response · Author response to Decision Letter 0]

18 Jul 2024

We thank the editor and reviewers for their recommendations and comments. All

questions and comments made by the reviewers and the editor were considered in the

"Responses to Reviewers" document.

---

## [Decision Letter · Decision Letter 1]

12 Aug 2024

PONE-D-23-33291R1At what cycle threshold level are dogs able to detect SARS-CoV-2 in humans?PLOS ONE

Dear Dr. Soler-Jimenez,

Thank you for submitting your manuscript to PLOS ONE. After careful consideration, we feel that it has merit but does not fully meet PLOS ONE’s publication criteria as it currently stands. Therefore, we invite you to submit a revised version of the manuscript that addresses the points raised during the review process.

We look forward to receiving your revised manuscript.

Kind regards,

Julian Ruiz-Saenz

Academic Editor

PLOS ONE

Reviewers' comments:

Reviewer's Responses to Questions

**Comments to the Author**

1. If the authors have adequately addressed your comments raised in a previous round of review and you feel that this manuscript is now acceptable for publication, you may indicate that here to bypass the “Comments to the Author” section, enter your conflict of interest statement in the “Confidential to Editor” section, and submit your "Accept" recommendation.

Reviewer #1: All comments have been addressed

Reviewer #2: All comments have been addressed

2. Is the manuscript technically sound, and do the data support the conclusions?

Reviewer #1: Yes

Reviewer #2: Partly

3. Has the statistical analysis been performed appropriately and rigorously? 

Reviewer #1: Yes

Reviewer #2: I Don't Know

4. Have the authors made all data underlying the findings in their manuscript fully available?

Reviewer #1: Yes

Reviewer #2: Yes

5. Is the manuscript presented in an intelligible fashion and written in standard English?

Reviewer #1: Yes

Reviewer #2: Yes

6. Review Comments to the Author

Reviewer #1: (No Response)

Reviewer #2: The SARS-CoV-2 virus by using trained dogs is still a huge challenge.

The authors should be addressing very carefully several aspects about their manuscript.

1.It is not clear what second generation of dogs mean? Are they F1?, Etc.

2.Each dog has specific characteristics, and they will have variability.

3.Did you expect to obtain near of 50% (6/11) of dogs with better results? It is a little bit high percentage.

4.The authors define some clinical characteristics of the patients, fever, pain, etc, could the authors to confirm mild or severe COVID-19 symptoms for the patients?

5.They should discuss if the dog gender (male vs female) has any relation with the detection?

6.In the manuscript should be stated if the dogs are detecting the viral genome or some volatile compounds associated to the infection. This is confused. Moreover, in some sections looks like the virus particle detection than genome.

7.Do the anecdotical experience is relevant?

8.They mention that dogs possess a significant advantage, because they can screen hundreds of people in a few hours on a daily basis, compared with the more expensive and logistically complex application of RADs. Is it accepted that a dog can screen hundreds of people in a few hours on daily basis?. This is a serious mistake, because a dog has only minutes to work not hours.

9.The authors are claiming that.. Although the dogs did not meet the sensitivity and specificity standards set by the WHO, their ability to perform large-scale screening in short periods of time makes them a valuable tool in the fight against the spread of COVID-19, especially in areas where RT-qPCR and rapid antigen testing are limited or expensive. This mean the probability will be the answer, this should be well discussed.

10.At present, COVID-19 is and will be another frequent viral disease, the discussion about more emergent viral diseases by RNA virus is very superficial. This a huge challenge.

11.The authors should discuss more about clinical and biological validation of the data.

12.It is a suggestion, to avoid first person in the manuscript.

13.To avoid using “can, might”, is much better “could”.

14.Thus, the abstract, discussion and conclusion should be improved.

15.Finally, To review the tables look like confused, and separate the data when the reports were done for viral antigen or genome detection.

7. PLOS authors have the option to publish the peer review history of their article (what does this mean?). If published, this will include your full peer review and any attached files.

Reviewer #1: No

Reviewer #2: No

---

## [Author Response · Author response to Decision Letter 1]

27 Sep 2024

We thank the editor and reviewers for their recommendations and comments. All

questions and comments made by the reviewers and the editor were considered in the

"Responses to Reviewers" document.

---

## [Decision Letter · Decision Letter 2]

11 Nov 2024

PONE-D-23-33291R2At what cycle threshold level are dogs able to detect SARS-CoV-2 in humans?PLOS ONE

Dear Dr. Soler-Jiménez,

Thank you for submitting your manuscript to PLOS ONE. After careful consideration, we feel that it has merit but does not fully meet PLOS ONE’s publication criteria as it currently stands. Therefore, we invite you to submit a revised version of the manuscript that addresses the points raised during the review process.

We look forward to receiving your revised manuscript.

Kind regards,

Julian Ruiz-Saenz

Academic Editor

PLOS ONE

Journal Requirements:

Reviewers' comments:

Reviewer's Responses to Questions

**Comments to the Author**

1. If the authors have adequately addressed your comments raised in a previous round of review and you feel that this manuscript is now acceptable for publication, you may indicate that here to bypass the “Comments to the Author” section, enter your conflict of interest statement in the “Confidential to Editor” section, and submit your "Accept" recommendation.

Reviewer #2: (No Response)

2. Is the manuscript technically sound, and do the data support the conclusions?

Reviewer #2: (No Response)

3. Has the statistical analysis been performed appropriately and rigorously? 

Reviewer #2: (No Response)

4. Have the authors made all data underlying the findings in their manuscript fully available?

Reviewer #2: (No Response)

5. Is the manuscript presented in an intelligible fashion and written in standard English?

Reviewer #2: (No Response)

6. Review Comments to the Author

Reviewer #2: This review is dedicated to authors Vidal-Martínez et al about, At what cycle threshold level are dogs able to detect SARS-CoV-2 in humans?.

Several aspects should be addressed and discussed in depth to improve the proper manuscript.

1. In the first generation 4 out 6 (66%) dogs marked when using 1 minute. For the second generation 5 out 11 (aprox. 45%) marked.

2. Is confused if all participants were RT-PCR tested. How many patients with Antigen positive were detected by the dogs? What about for the RT-PCR? And then to compare both data.

3. The authors should show, how was the detection limit and the number of viral copies calculus?

4. According to the symptoms, the dogs are for mild COVID-19, is this correct?

5. There is lack information about how the pseudo-scent was used, good or not for detection, could you please give some more information about the pseudo-scent? the sex dog influenced for the detection? Most of the dogs are male.

6. Who dogs were coccidiomycosis affected? How many rounds they recovered the detection

7 please to define 378 or 379 patients

Major comments

The introduction section is too long

The discussion section looks like results.

The authors should discuss the results. Not necessarily all the second-generation dogs should be better than the first one.

The abstract could be improved highlighting the results.

A better title could be considered.

It should be noted that there is a lack of information about the biochemical molecules involved, in other words, if there is a correlation (as you mention) between detection by dogs and N1/N2 viral proteins present in the patients.

Could you please explain the difference between 1 minute vs 5- or 10-minutes gauze exposed?

7. PLOS authors have the option to publish the peer review history of their article (what does this mean?). If published, this will include your full peer review and any attached files.

Reviewer #2: **Yes: **MAURICIO SALCEDO

---

## [Author Response · Author response to Decision Letter 2]

18 Dec 2024

We thank the editor and reviewers for their recommendations and comments. All questions and comments made by the reviewers and the editor were considered in the "Responses to Reviewers" document.

---

## [Editor Report · Decision Letter 3]

23 Dec 2024

At what cycle threshold level are dogs able to detect SARS-CoV-2 in humans?

PONE-D-23-33291R3

Dear Dr. Soler-Jiménez,

We’re pleased to inform you that your manuscript has been judged scientifically suitable for publication and will be formally accepted for publication once it meets all outstanding technical requirements.

Kind regards,

Julian Ruiz-Saenz

Academic Editor

PLOS ONE
---

## [Editor Report · Acceptance letter]

8 Jan 2025

PONE-D-23-33291R3 

PLOS ONE

Dear Dr. Soler-Jiménez, 

I'm pleased to inform you that your manuscript has been deemed suitable for publication in PLOS ONE. Congratulations! Your manuscript is now being handed over to our production team.

Kind regards, 

on behalf of

Dr. Julian Ruiz-Saenz 

Academic Editor

PLOS ONE